

# Regional Climate Model Evaluation System powered by Apache Open Climate Workbench v1.3.0: an enabling tool for facilitating regional climate studies

Huikyo Lee[1], Alexander Goodman[1], Lewis McGibbney[1], Duane Waliser[1], Jinwon Kim[2,3], Paul Loikith[4], Peter Gibson[1], and Elias Massoud[1]

[1]Jet Propulsion Laboratory, California Institute of Technology, Pasadena, USA
[2]Joint Institute for Regional Earth System Science & Engineering, University of California, Los Angeles, USA
[3]National Institute of Meteorological Sciences/Korean Meteorological Administration, Seogwipo, Korea
[4]Department of Geography, Portland State University, Portland, USA

**Correspondence:** Huikyo Lee (huikyo.lee@jpl.nasa.gov)

**Abstract.** The Regional Climate Model Evaluation System (RCMES) is an enabling tool of the National Aeronautics and Space Administration to support the United States National Climate Assessment. As a comprehensive system for evaluating climate models on regional and continental scales using observational datasets from a variety of sources, RCMES is designed to yield information on the performance of climate models and guide their improvement. Here we present a user-oriented document describing the latest version of RCMES, its development process and future plans for improvements. The main objective of RCMES is to facilitate the climate model evaluation process at regional scales. RCMES provides a framework for performing systematic evaluations of climate simulations, such as those from the Coordinated Regional Climate Downscaling Experiment (CORDEX), using in-situ observations as well as satellite and reanalysis data products. The main components of RCMES are: 1) a database of observations widely used for climate model evaluation, 2) various data loaders to import climate models and observations in different formats, 3) a versatile processor to subset and regrid the loaded datasets, 4) performance metrics designed to assess and quantify model skill, 5) plotting routines to visualize the performance metrics, 6) a toolkit for statistically downscaling climate model simulations, and 7) two installation packages to maximize convenience of users without Python skills. RCMES website is maintained up to date with brief explanation of these components. Although there are other open-source software (OSS) toolkits that facilitate analysis and evaluation of climate models, there is a need for climate scientists to participate in the development and customization of OSS to study regional climate change. To establish infrastructure and to ensure software sustainability, development of RCMES is an open, publicly accessible process enabled by leveraging the Apache Software Foundation's OSS library, Apache Open Climate Workbench (OCW). The OCW software that powers RCMES includes a Python OSS library for common climate model evaluation tasks as well as a set of user-friendly interfaces for quickly configuring a model evaluation task. OCW also allows users to build their own climate data analysis tools, such as the statistical downscaling toolkit provided as a part of RCMES.



# 1   Introduction

The anthropogenic climate change signal in the Earth system is not globally uniform. Instead, the magnitude and character of climate change, including long-term trends, year-to-year variability and characteristics of extremes of key meteorological variables, exhibit considerable geographical variability. For example, warming is of a larger magnitude in the polar regions
as compared with lower latitudes, due in part to a positive feedback related to rapidly receding polar ice caps (Gillett and Stott, 2009). This regional scale variability makes it an extremely difficult task to accurately make projections of climate change, especially on a regional scale. Yet characterizing present climate conditions and providing future climate projections at a regional scale are far more useful for supporting decisions and management plans intended to address impacts of climate change than global-scale climate change information.

Regional climate assessments heavily depend on numerical model projections of future climate simulated under enhanced greenhouse emissions that not only provide predictions of physical indicators but also indirectly inform on societal impacts, thus providing key resource for addressing adaptation and mitigation questions. These quantitative projections are based on Global and Regional Climate Models (GCMs and RCMs respectively). Because of the critical input such models have for decision makers, it is a high priority to make them subject to as much observational scrutiny as possible. This requires the systematic
application of observations, in the form of performance metrics and physical process diagnostics, from gridded satellite and reanalysis products as well as in-situ station networks. These observations then provide the target for model simulations, with confidence in model credibility boosted where models are able to reproduce the observed climate with reasonable fidelity. Enabling such observation-based, multivariate evaluations is needed for advancing model fidelity, performing quantitative model comparison, evaluation and uncertainty analyses, and judiciously constructing multi-model ensemble projections. These ca-
pabilities are all necessary to provide a reliable characterization of future climate that can lead to informed decision-making tailored to the characteristics of a given region's climate.

The Coupled Model Intercomparison Project (CMIP), currently in its fifth phase, is an internationally coordinated multi-GCM experiment that has been undertaken for decades to assess global-scale climate change. The Coordinated Regional Downscaling Experiment (CORDEX, Giorgi and Gutowski (2015); Gutowski et al. (2016)) is another modeling effort that
parallels CMIP but with a focus on regional-scale climate change. To complement CMIP, based on GCM simulations at relatively coarse resolutions, CORDEX aims to improve our understanding of climate variability and changes at regional scales by providing higher resolution RCM simulations for 14 domains around the world. Climate scientists analyze the datasets from CMIP and CORDEX, with the findings contributing to the IPCC assessment reports (e.g. IPCC (2013)). Plans and implementation for the next generation CMIP (CMIP6, Eyring et al. (2016a)) is now underway to feed into the next IPCC assessment
report (AR6, IPCC (2018)). In coordination with the IPCC efforts, the Earth System Grid Federation (ESGF) already hosts a massive amount of GCM output for past CMIPs, with CMIP6 and RCM output for CORDEX slated for hosting by ESGF as well. Because of the large variability across the models that contribute to CMIP, it is a high priority to evaluate the models systematically against observational data, particularly from Earth remote sensing satellites (e.g., Teixeira et al. (2014); Freedman et al. (2014); Stanfield et al. (2014); Dolinar et al. (2015); Yuan and Quiring (2017)). As more GCMs and RCMs participate



in the two projects, the ESGF infrastructure faces a challenge of providing a common framework where users can analyze and evaluate the models using the observational datasets hosted on the ESGF, such as the observations for Model Intercomparison Projects (obs4MIPs; Ferraro et al. (2015); Teixeira et al. (2014)) and reanalysis data (ana4MIPs, ana4MIPs (2018)).

As careful and systematic model evaluation is widely recognized as critical to improving our understanding of future climate
change, there have been other efforts to facilitate this type of study. Here we briefly describing existing model evaluation toolkits for CMIP GCMs. The Community Data Analysis Tools (CDAT, LLNL (2018c)) is a suite of software that enables climate researchers to solve their data analysis and visualization challenges. CDAT has already successfully supported climate model evaluation activities for DOE's climate applications and projects /citepclimatemodeling, such as the IPCC AR5 and Accelerated Climate Modeling for Energy. The Earth System Model Evaluation Tool (ESMValTool, Eyring et al. (2016b); Lauer et al.
(2017)) is another software package that offers a variety of tools to evaluate the CMIP GCMs. The diagnostic tools included in ESMValTool are useful not only for assessing errors in climate simulations but also providing better understanding of important processes related to the errors. While ESMValTool has facilitated global-scale assessments of the CMIP-participating GCMS, the development and application of infrastructure for a systematic, observation-based evaluation of regional climate variability and change in RCMs is relatively immature, owing in considerable part to the lack of a software development platform to
support climate scientist users around the world.

The main advantage of RCMs, with their limited spatial domains, over GCMs is their higher spatial resolution. A number of previous studies have demonstrated the value of using RCMs with higher horizontal resolutions than GCMs in projecting future climate changes at regional scales (e.g. Lee and Hong (2014); Lee et al. (2017); Wang et al. (2015); Di Luca et al. (2012, 2016); Diaconescu and Laprise (2013); Poan et al. (2018)), which are attained both by improved accuracy of topographic presentation
and also more explicit numerical computation of dynamical and physical processes as based on first principles.

Therefore, it is crucial to leverage the added value of RCMs, because they will improve our estimation of the regional impacts of climate change. RCM experiments, such as CORDEX, will play a critical role in providing finer scale climate simulations. This role also fits the U.S. National Climate Assessment's (NCA, Jacobs (2016)) strategic objective, to produce a quantitative national assessment with consideration of uncertainty. Here, assessment of the uncertainty in simulated climate
requires comprehensive evaluation of many RCMs against in-situ and remote sensing observations and regional reanalysis data products. The observation-based evaluation of multiple RCMs with relatively high resolution also requires the appropriate architectural framework capable of manipulating large datasets for specific regions of interest.

Recognizing the need for an evaluation framework for high-resolution climate models with a special emphasis on regional scales, the Jet Propulsion Laboratory (JPL) and the Joint Institute for Regional Earth System Science and Engineering
(JIFRESSE) at the University of California, Los Angeles (UCLA) have developed a comprehensive suite of software resources to standardize and streamline the process of interacting with observational data and climate model output to conduct climate model evaluations. The Regional Climate Model Evaluation System (RCMES, JPL (2018a); Mattmann et al. (2014); White-hall et al. (2012)) is designed to provide a complete start-to-finish workflow to evaluate multi-scale climate models using observational data from the RCMES database and other data sources including the ESGF (e.g. obs4MIPs), JPL's Physical
Oceanographic Data Active Archive Center (PO.DAAC, citetpodaac) and any OPeNDAP (OPeNDAP, 2018) server.



RCMES is mutually complementary to CDAT and ESMValTool with the main target of supporting CORDEX and NCA communities by fostering collaboration of climate scientists in studying climate change at regional scales. To promote greater collaboration and participation of the climate research community within the RCMES development process, we transitioned from a closed-source development process to an open-source software (OSS) community driven project hosted in the public forum and therefore subject to public peer review, something which has significantly improved the overall project quality and standards the community and project holds itself to. RCMES is Python-based OSS powered by the Apache Software Foundation's (ASF) Open Climate Workbench (OCW) project. OCW is a python library for common model evaluation tasks (e.g. data loader, subsetting, regridding, and performance metrics calculation) as well as a set of user-friendly interfaces for quickly configuring a large-scale regional model evaluation task. OCW acts as the baseline infrastructure of RCMES, allowing users to build their own climate data analysis tools and workflows.

The primary goal of this paper is to document RCMES as powered by OCW, as well as describe the workflow of evaluating RCMs against observations using RCMES. Recent developments on the workflow include a template for performing systematic evaluations of CORDEX simulations for multiple variables and domains. We also demonstrate the benefits of developing RCMES in a collaborative manner by leveraging ASF's OCW project. Experience tells us that there is strong demand for peer-reviewed documentation in support of RCMES used by climate scientists, and this paper provides exactly that.

The paper is organized as follows. Section 2 describes the overall software architecture of RCMES, followed by detailed description on each component of RCMES in Section 3. Section 4 presents the value of developing OSS within a public community driven model. Section 5 provides summary and future development plans.

## 2 Overall structure of RCMES

RCMES provides datasets and tools to assess the quantitative strengths and weakness of climate models, typically under present climate conditions for which we have observations for comparison, which then forms a basis to quantify our understanding of model uncertainties in future projections. The system and workflow of RCMES are schematically illustrated in Figure 1. There are two main components of RCMES. The first is a database of observations, and the second is the RCMES toolkits. The workflow of climate model evaluation implemented by RCMES starts with loading observation and model data. Currently, RCMES users can load datasets from three different sources: 1) RCMES database, 2) local storage, 3) ESGF servers (e.g. obs4MIPs), and any combinations of 1)-3). Access to other datasets archived on remote servers will be tested and implemented in future versions. Once the datasets are loaded, RCMES subsets the datasets spatially and temporally, and optionally regrids the subsetted datasets, compares the regridded datasets, calculates model performance metrics, and visualizes/plots the metrics. The processed observational and model datasets are saved in a netCDF file. All of this model evaluation process is controlled by user input. Because RCMES captures the entire workflow, another user can reproduce the same results using the captured workflow.

Figure 2 displays the step-by-step pathway for learning and using RCMES. As an introduction to RCMES, the simple, but intuitive Command Line Interface (CLI) is provided. Running RCMES using a configuration file (CFiles) enables a basic, but



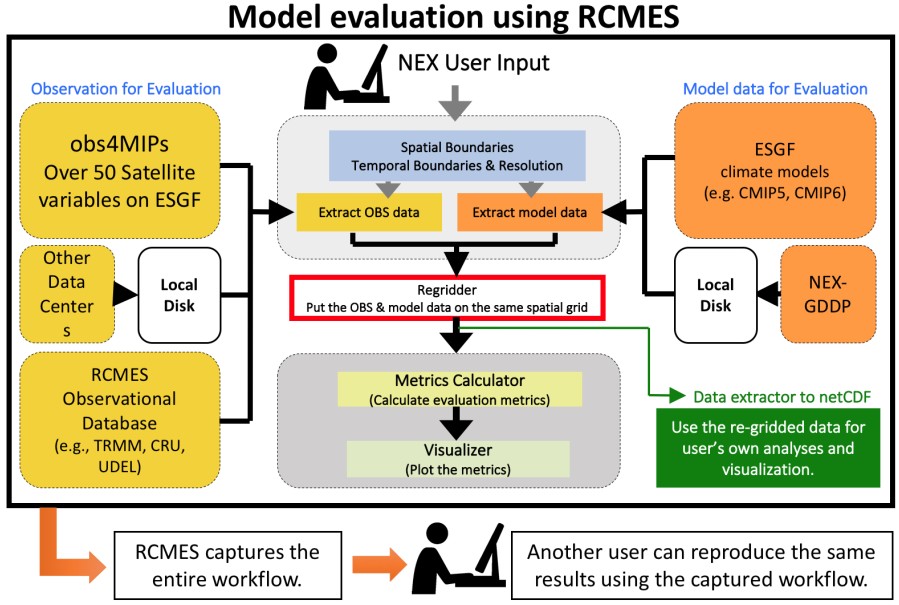

**Figure 1.** The outline and data flows within RCMES (adapted from JPL (2018a)).

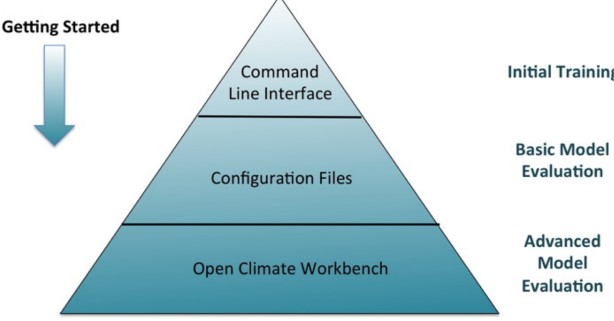

**Figure 2.** The approach to getting acquainted and using RCMES and OCW (adapted from JPL (2018a)).

important and comprehensive evaluation of multiple climate models using observations from various sources. Advanced users can utilize Open Climate Workbench library to build up scripts for customized data analysis and model evaluation.

The CLI example included in the RCMES package compares annual precipitation over the contiguous United States in the year 2000 between CRU and the WRFG RCM (Figure 3). A step-by-step online tutorial to run this example can be found on the RCMES website (JPL, 2018b). The CLI requires users to select an option from a numbered list in each step of the RCM evaluation. Although the model evaluation through CLI is limited to calculating climatological biases of one RCM simulation against one of the observational datasets from the RCMES database, the CLI offers RCMES users an opportunity to learn




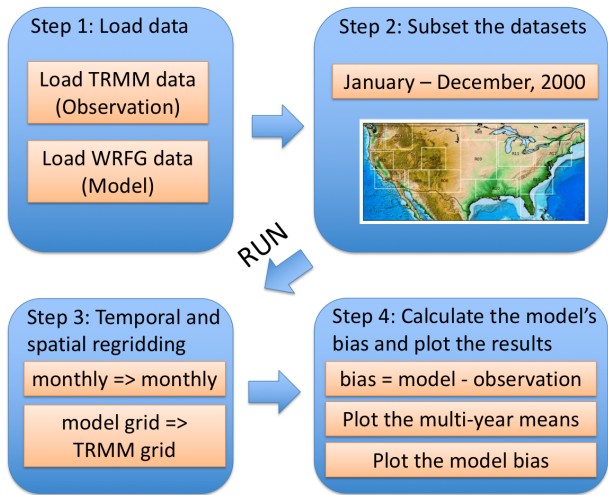

**Figure 3.** The four steps of model evaluation included as a CLI example of RCMES.

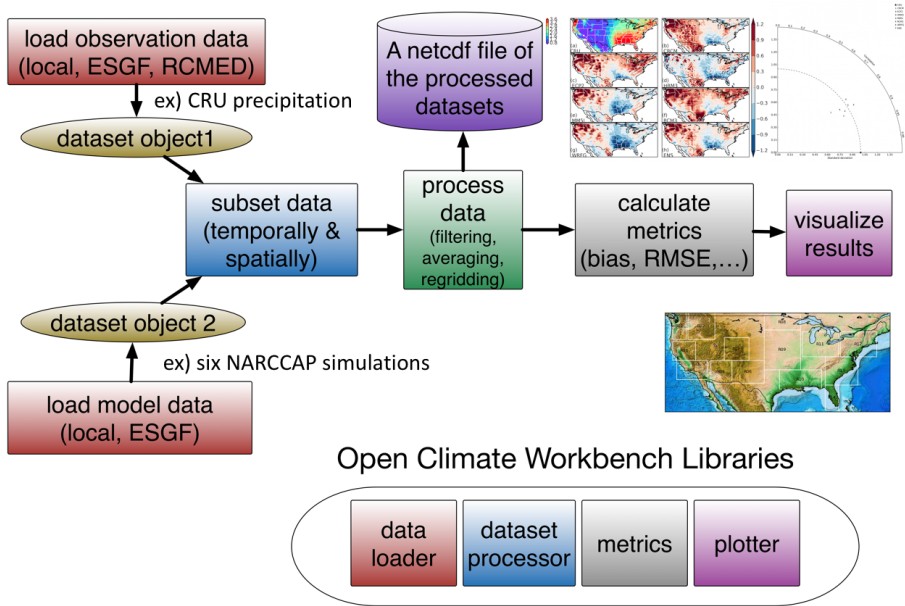

**Figure 4.** The process of evaluating climate models using datasets from multiple sources and modularized OCW libraries. The evaluation metrics and plots in Kim et al. (2013) are shown as an example.

the basic process of model evaluation, which includes inputting model datasets and observations, specifying gridding options, calculating the model's bias and plotting the results.





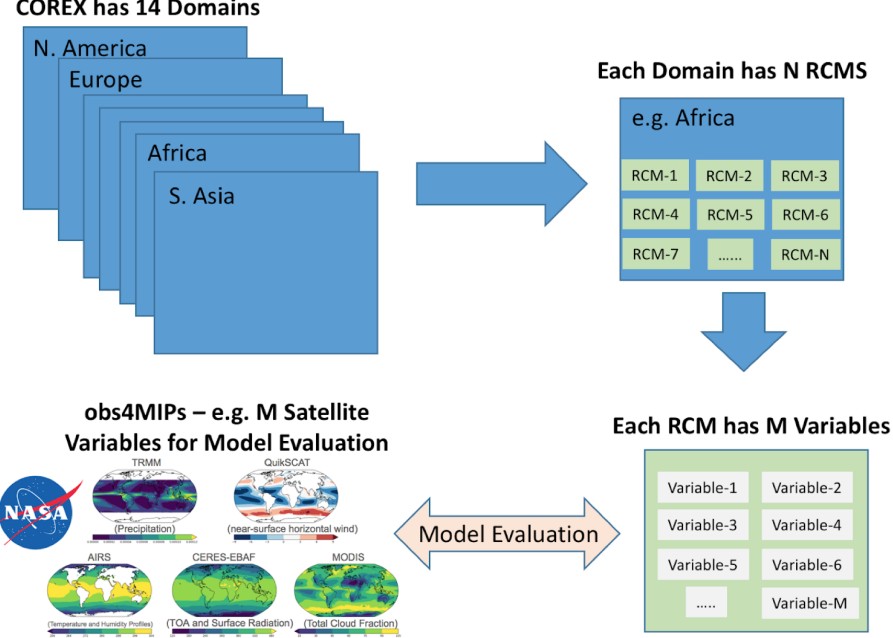

**Figure 5.** The multi-model, multi-variable evaluation against satellite observations from obs4MIPs over the 14 CORDEX domains.

Configuration files (CFiles) lie at the heart of RCMES to support user customized evaluation of multiple RCMs, such as those participating in CORDEX, and their ensemble means. To maximize their utility, RCMES CFiles use the Python yaml format of namelist files that are familiar to many climate scientists who run climate models. CFiles include a complete start-to-finish workflow of model evaluation as shown in Figure 4. In CFiles, users can define the evaluation domain, the time period of evaluation, regridding options, and performance metrics to calculate. Released RCMES packages include example CFiles to reproduce the plots/diagrams in the four selected peer-reviewed journal articles including (Kim et al., 2013, 2014). Kim et al. (2013) and Kim et al. (2014) evaluate RCM simulations over the North America and Africa respectively. The tutorials on the RCMES websites (JPL, 2018b) provide step-by-step instructions to reproduce all of the figures included in the two published articles.

Figure 5 illustrates the latest RCMES development to provide an easy solution to evaluate key variables simulated by CORDEX RCMs against satellite observational datasets from obs4MIPs. Currently, running RCMES with a CFile enables evaluation of multiple models for one variable over a specified domain. Given more than 30 different obs4MIPs variables, RCMES provides a script named 'cordex.py' to generate configuration files automatically. Users are requested to select one of the fourteen CORDEX domains and provide a directory path on a local machine where obs4MIPs and model datasets are archived. Then, the script extracts variable and CORDEX domain information from searched file names in subdirectories of the given directory path by utilizing the climate and forecast (CF) naming convention (CF, 2018) for the obs4MIPs and CORDEX data files available from ESGF servers. The RCMES website also provides these examples of the multi-model, multi-variable





evaluation for several CORDEX domains (North America, Europe, and Africa) as a part of the RCMES tutorial. As an example, running RCMES for CORDEX North America domain with 12 variables and 3 seasons (36 unique evaluations with 5 datasets each) takes about 45 minutes on a multi-core Linux computing platform.

Continuously reflecting climate scientists' needs and facilitating greater scientific yield from model and observation datasets are important for expanding the future user-base of RCMES. Despite the development environment encouraging participation of open communities, one of the main challenges in using RCMES for evaluating climate models has been to support dataset files in various formats. To address this issue, the development of flexible and versatile data loaders that read files with different formats is required. Another limiting factor of the CFile-based format utilized by RCMES is that it is not easy to calculate sophisticated diagnostics with which the model evaluation process does not fit into the workflow in Figure 4. Most of the model evaluation metrics included in the current RCMES distribution are intuitive but relatively rudimentary, such as a bias, a root mean square error, and linear regression coefficients. Although climate scientists obtain an insight into climate models with these basic metrics calculated and visualized with RCMES, model assessments and, ultimately, future model improvement require more comprehensive and mathematically rigorous metrics for quantifying models' uncertainty.

Note that RCMES has used the OCW library to build individual components. To meet the dynamic requirements of RCMES users, the OCW package provides several advanced analysis examples of model evaluation by combining various modules from the OCW python library that can be executed independently. Providing a suite of interchangeable modules and functions that implement analysis of observational and model datasets is more beneficial to the climate science community than developing RCMES in a more complicated way by adding more CFile examples. For example, users can use the OCW file loaders and data processors to obtain an intermediary output netCDF file. The file can be used with the users' own script written in any other programming language. It is also possible to mix OCW file loaders and processors with other Python libraries. The tutorial page on the RCMES website also describe various applications of using OCW modules for advanced analyses of climate science data. For instance, Kim et al. (2017) and citetLee17 use RCMES to compare the high-resolution simulations made using NASA-Unified Weather Research and Forecasting model (Peters-Lidard et al., 2015) with daily and hourly observations. The tutorials on the RCMES websites provide step-by-step instructions to reproduce figures included in these published articles.

## 3   Components of RCMES

In the following, we describe seven software components of RCMES, 1) data loader, 2) the RCMES database, 3) dataset processor, 4) metrics, and 5) plotter, 6) statistical downscaling module, and 7) installation package options for disseminating RCMES. Our website (JPL, 2018a) updated with new developments and examples on a regular basis is also a vital component of RCMES. As illustrated in Figure 4, climate model evaluation using RCMES starts with loading observation and model data using OCW. The observation data can be pulled from different sources. The main function of the dataset processor is to subset and regrid the dataset objects. The dataset processor also saves the processed datasets in a user-specified netCDF file. Since individual modules in the data loader and dataset processor can be used and combined for various purposes, we provide a user-friendly manual in the current manuscript describing the modules.



## 3.1 Data loader

The first step in performing any climate model evaluation is determining which observational and model datasets to use and retrieving them for subsequent use. Ideally, one would like to support a standardized, user-friendly interface for data retrieval from the most common sources used by climate scientists. OCW facilitates this by providing RCMES with several dataset

loaders to read and write CF-compliant netCDF files, and loaders for specific datasets. The objective of offering some specific loaders, such as loaders for the Weather Research and Forecasting (WRF, Skamarock et al. (2008)) model's raw output or the Global Precipitation Measurement (GPM, Huffman et al. (2015)) observation data, is to expand the convenience of users' customized model evaluation studies using observation and model data files from various sources without file conversion. By design, all of the dataset loaders return a Dataset object or a list of multiple Dataset objects which store gridded variables

with arrays of latitudes, longitudes, times, and variable units regardless of the format of the original data files. When handling input gridded data in three spatial dimensions with elevation, users need to specify elevation_index, an optional parameter of dataset loaders. By default, elevation_index is zero. The following subsection describe four data sources for which RCMES has built-in support.

### 3.1.1 RCMES Database (RCMED)

RCMES is a comprehensive system whose distribution includes its own database of observational datasets that can be readily accessed for evaluating climate models. Currently, the database provides 14 datasets from ground stations and satellites. Among those, precipitation data from NASA's Tropical Rainfall Measuring Mission (TRMM, Huffman et al. (2007)), temperature and precipitation data from the Climate Research Unit (CRU, Harris et al. (2014)), are widely used by the climate science community. RCMED also provides evaporation, precipitation, and snow water equivalent datasets from NASA's reanalysis

products. The RCMED loader requires the following parameters:

- dataset_id, parameter_id: Identifiers for the dataset and variable of interest (https://rcmes.jpl.nasa.gov/content/data-rcmes-database)

- min_lat, max_lat, min_lon, max_lon, start_time, end_time: Spatial and temporal boundaries used to subset the dataset domain.

From an implementation perspective, these parameters are used to format a Representational State Transfer (REST) query which is then used to search the RCMED server for the requested data. The loaders provided by OCW for two of the other data sources (ESGF and PO.DAAC) also work in a similar fashion.

### 3.1.2 Local filesystem

The simplest and most standard way to access Earth science datasets is storing netcdf files in the local filesystem. The

ocw.data_source.local module reads, modifies and writes to the locally stored files. In addition to loading one Dataset object from one file, this module also contains loaders for loading a single dataset spread across multiple files, as well as multiple





datasets from multiple files. In each case, dataset variables and netCDF attributes are extracted into OCW Datasets using the netCDF4 or hdf5 python libraries. Most of the remote data sources described in the next few sessions also depend on these loaders, since they generally entail downloading datasets to the local filesystem as the first step, then loading them as locally stored files. The following parameters are required to load one file:

– file_path: The absolute or relative file path to the netCDF file of interest.

     – variable_name: The name of the variable to load, as is defined in the netCDF file.

By default, the local loader reads the spatial and temporal variables (latitude, longitude, and time) by assuming they are commonly used variable names (e.g., lat, lats, or latitude), which should typically be the case if the file to be loaded is CF-compliant. However, these variable names can be manually provided to the loader for files with more unusual naming conventions for these

variables. In the loader, these parameters are respectively lat_name, lon_name, and time_name.

### 3.1.3    Earth System Grid Federation (ESGF)

The Earth System Grid Federation (ESGF), led by Program for Climate Model Diagnosis and Intercomparison (PCMDI) at LLNL (Taylor et al., 2012), provides CF-compliant climate datasets for a wide variety of projects including CMIP, obs4MIPS, and CORDEX. Provided that the user can authenticate with a registered account (via OpenID), data can be readily accessed

through a formatted search query in a similar manner to the PO.DAAC and RCMED data sources as the first step of model evaluation using RCMES. The ESGF loader requires the following parameters:

     – dataset_id: Identifier for the dataset on ESGF.

     – variable_name: The name of the variable to select from the dataset, in CF short name form.

     – esgf_username, esgf_password: ESGF username (e.g., OpenID) and password used to authenticate.

ESGF provides its data across different nodes which are maintained by a variety of climate research and modeling institutes throughout the world. The loader searches the JPL node by default, which contains CMIP5 and obs4MIPS data. However, if datasets from other projects are desired, then the base search URL must also be specified in the loader via the search_url parameter. For example, the DKRZ node (search_url=https://esgf-data.dkrz.de/esg-search/search) should be used if the user wishes to obtain CORDEX model output.

### 3.1.4    NASA's Physical Oceanographic Data Active Archive Center (PO.DAAC)

The Earth Observing System Data and Information System (EOSDIS) is a key core capability in NASA's Earth Science Data Systems Program. It provides end-to-end capabilities for managing NASA's Earth science data from various sources: satellites, aircraft, field measurements, and various other programs, mainly targeting physical oceanography related variables. In OCW, we have implemented data source support for JPL's PO.DAAC, one of NASA's 12 major Distributed Active Archive



Centers (DAACs) which are critical components of EOSDIS which are located throughout the United States. Current functionality includes the ability to retrieve and extract full granules and/or specific variables from over 50 Level 4 blended datasets covering myriad of gridded spatial resolutions between 0.05 and 0.25 degrees, a range of temporal resolutions from daily to monthly, parameters (Gravity, Glaciers, Ice Sheets, Ocean Circulation, Ocean Temperature, etc.), latencies (Near Real

Time, Delayed Mode and Non-Active), collections (Cross-Calibrated Multi-Platform Ocean Surface Wind Vector Analysis Fields, Climate Data Record, Group for High Resolution Sea Surface Temperature (GHRSST), etc.), platforms (ADEOS-II, AQUA, AQUARIUS_SAC-D, Coriolis, DMSP-F08, etc.), sensors (AATSR, AHI, AMR, AMSR-E, AMSR2, AQUAR-IUS_RADIOMETER, etc.), spatial coverages (Antarctica, Atlantic and Pacific, Baltic Sea, Eastern Pacific Ocean, Global, etc.) and data formats (ASCII, NETCDF and RAW).

As all data loaded by RCMES loaders generate Dataset objects with spatial grid information, currently only Level 4 blended PO.DAAC datasets are suitable for evaluating climate models using RCMES. To synchronize the dataset search and selection, the PO.DAAC loader provides a convenience utility function which returns an up-to-date list of available level 4 granule dataset IDs which can be used in the granule extraction process. Once the list_available_level4_extract_granule_dataset_ids() function has been executed and a suitable dataset_id selected from the returned list, the PO.DAAC loader can be invoked with the

following granule subset and granule download functions.

subset_granule:

- – variable: The name of the variable to read from the dataset.

- – name=": (Optional) A name for the loaded dataset.

- – path='/tmp', (Optional) a path on the filesystem to store the granule.

– input_file_path=", Path to a json file which contains the subset request that you want to send to PO.DAAC. The JSON syntax is explained at https://podaac.jpl.nasa.gov/ws/subset/granule/index.html

extract_l4_granule:

- – variable: The name of the variable to read from the dataset.

- – dataset_id=": dataset persistent ID. The ID is required for a granule search. Example: PODAAC-CCF35-01AD5

– name=", (Optional) A name for the loaded dataset.

- – path='/tmp', a path on the filesystem to store the granule.

## 3.2  Dataset processor

Once the data is loaded, the next step is to homogenize the observational and model datasets such that they can be compared with one another. This step is necessary because in many cases, the input datasets can vary in both spatial and temporal

resolution, domain, and even physical units. Operations for performing this processing step on individual OCW datasets can





be found in the dataset_processor module, which will be described in greater detail in the following subsections. All of these data processing tools make use of the numpy and scipy libraries (van der Walt et al., 2011) of Python.

### 3.2.1 Subsetting

The first processing step is subsetting, both in space and time. This is especially important for evaluating RCMs since many observational datasets are defined on global grids, so a simple subset operation can greatly reduce the potential memory burden. The following parameters are required for subsetting:

– subregion: The target domain. This includes spatial and temporal bounds information, and can be derived from a rectangular bounding box (lat, lon pairs for each corner), one of the fourteen CORDEX domain names (e.g., North America), or a mask matching the dimensions of the input dataset.

For clearer tractability, a subregion_name parameter can be provided to label the domain. If the user wishes to subset the data further based on a matching value criterion, these may be provided as a list via the user_mask_values parameter. Finally, the extract parameter can be toggled to control whether or not the subset is extracted (e.g., the output dimensions conform to the given domain) or not (the original dimensions of the input dataset are preserved, and values outside the domain are masked).

A temporal (or rather, seasonal) subset operation is also supported. The parameters are:

– month_start, month_end: Start and end months which denote a continuous season. Continuous seasons which cross over new years are supported (e.g., (12, 2) for DFJ).

The average_each_year parameter may be provided to average the input data along the season in each year.

### 3.2.2 Temporal resampling and spatial regridding

Having addressed inhomogeneities in the input datasets with respect to domain, discrepancies in spatial and temporal resolution need to be considered. Resampling data to a lower temporal resolution (e.g., from monthly to annually) is performed via a simple arithmetic mean and requires the following parameter:

– temporal_resolution: The new temporal resolution, which can be annual, monthly, or daily.

Spatial regridding, on the other hand, is the most computationally expensive operation. OCW provides a relatively basic implementation which utilizes SciPy's griddata routine for bilinear interpolation of 2D fields. The parameters required by the user are:

– new_latitudes, new_longitudes: one or two-dimensional arrays of latitude and longitude values which define the grid points of the output grid.

### 3.2.3 Unit conversion

Because physical variables can be expressed in a large variety of units, RCMES supports automatic unit conversion for a limited subset of units. These include conversion of all temperature units into Kelvin and precipitation into mm/day.





### 3.3 Metrics and plotter

The current RCMES distribution offers various model evaluation metrics from ocw.metrics. There are two different types of metrics. RCMES provides basic metrics, such as bias calculation, Taylor diagram, and comparison of time series. Earlier RCMES publications (Kim et al., 2013, 2014) show how to use the basic metrics in multi-model evaluation as illustrated in

Figure 4. The metrics module also provides more advanced metrics. For example, a joint histogram of precipitation intensity and duration (Kendon et al., 2014; Lee et al., 2017) can be calculated using hourly precipitation data from observations or model simulations. The joint histogram can be built for any pairs of two different variables as well. Another set of metrics that are recently added enables evaluation of long-term trends simulated climate models over the contiguous United States and analysis of the associated uncertainty.

The final step in any model evaluation is visualizing calculated metrics. For this purpose, OCW includes some utilities for quickly generating basic plots which make use of the popular matplotlib plotting package (Hunter, 2007). These include: time series, contour maps, portrait diagrams, bar graphs, and Taylor diagrams. All of the included plotting routines support automatic subplot layout, which is particularly useful when RCMES users evaluate a large number of models all together.

### 3.4 Statistical downscaling using RCMES

As stated in the introduction, the spatial resolution of GCMs is typically coarse relative to RCMs due to the high computational expense of GCMs. To use output from GCM simulations for studying regional climate and assessing impacts, GCM simulations typically need to be downscaled, a process that generates higher resolution climate information from lower resolution datasets. Although RCMs provide a physics-driven way to dynamically downscale GCM simulations, the computational expense of running RCMs can be substantial. In addition, it is sometimes necessary to correct the errors in the simulated climate that are

deviated from observations.

Recognizing the needs for downscaling and error correction, RCMES provides a toolkit for statistically downscaling simulation output from CMIP GCMs and correcting the output for present and future climate. The statistical downscaling toolkit supports different methods adapted from previous studies including a simple bias correction, bias correction and spatial disaggregation, quantile mapping, and asynchronous regression approach (Stoner et al., 2013). All of these simple statistical

downscaling approaches are intuitive and easy to understand.

The statistical downscaling script accepts users' input from a CFile. This CFile is somewhat different from one for model evaluation using RCMES but uses the same yaml format. The input parameters include a geographical location of the point to downscale GCM output, temporal range, and source of observational and model datasets. Users can select one of observations from RCMED (Section 4.1.1) or local file system (Section 4.1.2). GCM output needs to be stored in the local file system. The

statistical downscaling generates 1) a map file, 2) histograms, and 3) a spreadsheet. The map shows a location of downscaling target specified in the CFile. The histograms using numbers in the spreadsheet show distributions of the observation and the original and downscaled model datasets. The RCMES tutorial explains the entire process of statistical downscaling in great detail.



### 3.5 Download and installation of RCMES

There are two ways to download and install RCMES. One is executing a one-line command to install one of the RCMES packages using the terminal application. The other is with a virtual machine (VM) environment. The latter is slower than the package installation, but it provides Linux OS and all required python libraries that can run on any types of users' computers.

5 #### 3.5.1 Installation package

RCMES users can install OCW into their python environments with a single command using PyPI (pip install ocw) or Conda (conda install ocw). The latter is the recommended installation method, as it can automatically handle dependency management for a wide variety of platforms including Windows, OS X, and Linux x86-64. The package binaries are officially hosted on conda-forge (CONDA-FORGE, 2018), a community managed package distribution channel which builds package binaries for 10 each release using a variety of popular continuous integration services on Github. This approach makes it easy to support a large matrix of python versions and platforms.

#### 3.5.2 Virtual machine image

For novice users who are not familiar with Unix and Linux terminals, there is an option to run RCMES without the package installation. As a completely self-contained package running in a virtual host container, the virtual machine (VM) approach 15 offers a plug and play approach that enables users to quickly begin exploring RCMES without any trouble in the installation process. The RCMES VM image contains Linux, the latest Python, OCW libraries, dependencies, data analysis examples, as well as the latest version of sample datasets to execute some tutorials.

The VM image is an implementation of the same computer environment as the OCW developers with users' own computers. Several RCMES training sessions have utilized the VM image as a means for sharing RCMES source codes and generating the 20 tutorial examples with user specific customization. As an additional benefit, when a user no longer requires the VM image, the image can be easily replaced or completely removed from the host environment without affecting other software or libraries that may be installed.

### 4 Community software development

In the early development phase of RCMES, we encountered climate scientists from around the world who have the desire to 25 publish their source codes for climate data analysis and model evaluation, proactively contributing to the OSS and have their code made available as official software releases protected by appropriate software license(s). These activities require extra effort on refactoring, testing, and documenting their codes as well as for them to become subject to peer review. However, it is not easy to find a sustained collaborative platform where climate scientists can spontaneously share their software updates with others.



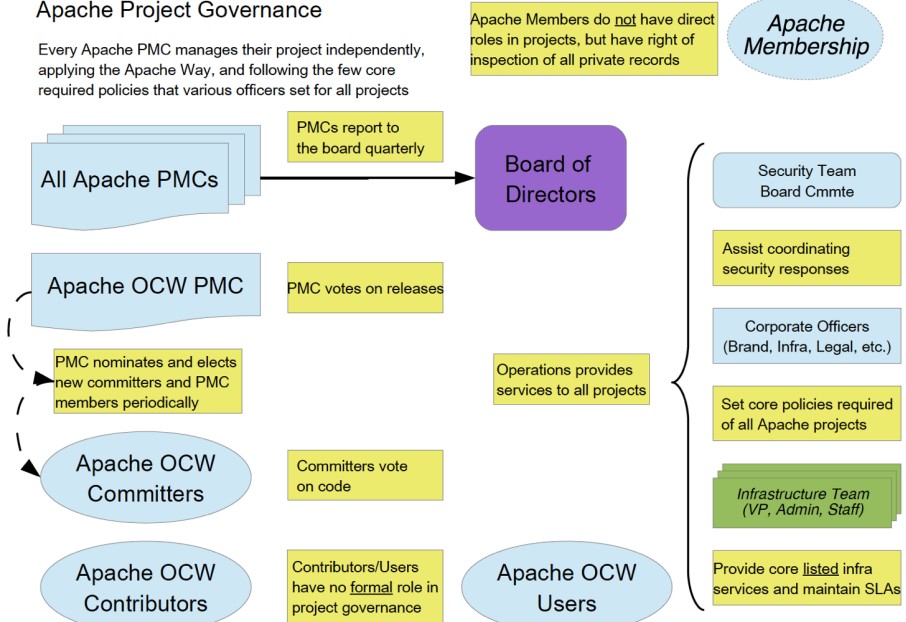

**Figure 6.** Schematic representation of Apache OCW project governance.

To encourage the potential benefit of collaborative development of OSS, the RCMES team decided to transition the development process from a closed-source project at JPL to an open-source, community-driven project hosted by the Apache Software Foundation (ASF, citetapache) in 2013. The goal of this transition was to not only make the RCMES codebase readily available under a permissive license e.g. the Apache Licence v2.0 (ASF, 2018b), but also to focus on growing this sustainable and healthy

5  environment, named as OCW. Any climate scientists can make contributions to OCW that comprises RCMES. At this time, the opinion was that, without an active and passionate community, RCMES becomes of lesser value to everyone and anyone. Hence, the OCW project mantra has always followed a 'community over code' model. This aspiration would eventually result in OCW becoming the second ever JPL-led project to formally enter incubation at the ASF, further establishing JPL as a leader in well governed, sustainable and successful transition of Government-funded software artifacts to the highly recognized and

10  renowned ASF where the primary goal is to provide software for the public good. Having undertaken an incubation period (ASF, 2018a) of about 18 months, OCW successfully graduated from the Apache Incubator in March 2014 with the result being that OCW could stand alongside 100 top level software projects all governed and developed under the iconic Apache brand. Since entering incubation, the OCW Project Management Committee (PMC) has developed, managed and successfully undertaken no fewer than 9 official Apache releases, a process involving stringent review by the extensive, globally distributed

15  Apache community (some 620 individual Members, 5,500 Committers and many thousands more contributors). In addition, over the years OCW has been presented in countless conferences.





From the outset, the OCW project development has followed a well-established, structured and independently managed community development process where a governing body, referred to as the OCW PMC, is responsible for the proper management and oversight of OCW, voting on software product releases and electing new PMC members and committers to the project. As well as deciding upon and guiding the project direction, the OCW PMC reports directly to the Apache Software Foundation board of directors on a quarterly basis providing updates on community and project specifics such as community activity, project health and any concerns, last release(s) additions to the PMC, etc. Figure 6 provides an overview of the project governance also providing additional context as to where and how OCW fits into the foundational structure of the ASF.

OCW follows a review-then-commit source code review process where new source code contributions from any party is reviewed by typically more than one OCW committer who has write permissions for the OCW source code. The OCW community use several tools which enable the project and community to function effectively and grow. These include tailored instances of software project management tools such as

– Wiki Documentation; Confluence wiki software enabling the community to provide documentation for release procedures, functionality, software development process etc.

– Source Code Management; the canonical OCW source code is hosted by the ASF in Git, with a public synchronized mirror provided at https://github.com/apache/climate

– Issue Tracking; OCW committers use the popular JIRA issue tracking software for tracking development including bug fixes, new features and improvements, development drives and release planning.

– Build Management and Continuous Integration; both Jenkins and TravisCI provide nightly builds and pull request continuous integration test functionality respectively ensuring that the OCW test suite passes and that regressions are mitigated between releases.

– Project Website; the primary resource for OCW is the project Website available at https://climate.apache.org. This is hosted by the ASF and maintained by the OCW PMC.

Using the tools listed above, OCW and accompanying RCMES have been released by a release manager who is also a Committer and PMC member with write access to the OCW source code. OCW release candidates follow strict peer review, meaning that they are official Apache releases (e.g. one which has been endorsed as an act of the Foundation by the OCW PMC). The OCW PMC must obey the ASF requirements on approving any release meaning that a community voting process needs to take place before any OCW release can officially be made. For a release vote to pass, a minimum of three positive votes and more positive than negative votes must be cast by the OCW community. Releases may not be vetoed. Votes cast by PMC members are binding. Before casting +1 binding votes, individuals are required to download all signed source code packages onto their own hardware, verify that they meet all requirements of ASF policy on releases, validate all cryptographic signatures, compile as provided, and test the result on their own platform. A release manager should ensure that the community has a 72-hour window of opportunity to review the release candidate before resolving or cancelling the release process.





All of the OCW source codes and RCMES are freely available under the Apache License v2.0 which is not only a requirement for all Apache projects but also desired by the OCW community due to the core project goals of open, free use of high quality OSS. Over time we have witnessed a significant degree of community growth through availability of OCW through an open, permissive license such as the Apache License v2.0.

## 5   Summary and future development plans

Although there are other open-source software toolkits that facilitate analysis and evaluation of climate models, there is a need for climate scientists to participate in the development and customization of software to study regional climate change. To meet the need, RCMES provides tools to analyze and document the quantitative strengths and weakness of climate models to quantify and improve our understanding of uncertainties in future predictions. The model evaluation using RCMES includes
loading observational and model datasets from various sources, a processing step to collocate the models and observations for comparison, and steps to perform analysis and visualization.

We have encouraged community participation in developing RCMES by releasing the database and software toolkits as open source under the Apache Software License version 2. Our experience with RCMES development has shown us that open source software is a means for ensuring sustained innovation and development of RCMES, and a pathway for facilitating informed
decisions regarding climate change at a regional scale.

The present version of RCMES is populated with a number of contemporary climate and regionally relevant satellite, re-analysis and in-situ-based datasets, with the ability to ingest additional data sets of wide variety of forms, ingest climate simulations, apply a number of useful model performance metrics and visualize the results. However, at present, the regridding routines included in the RCMES distribution are rudimentary, such as bilinear and cubic spline interpolation. Therefore,
our future RCMES development will prioritize advancing the regridding scheme in OCW's dataset processor including a tool for remapping datasets with different spatial resolutions into the Hierarchical Equal Area isoLatitude Pixelization (HEALPix, Gorski et al. (2005)) grids. HEALPix is an open-source library for fast and robust multi-resolution analysis of observational and model datasets regridded into HEALPix pixels, which have been widely used by astronomers and planetary scientists.

Future RCMES development will also include new metrics for the calculation and interrogation of rainfall extremes in in-situ
observations, satellite and regional climate model simulations. These will include a suite of precipitation metrics based on the Expert Team on Climate Change Detection and Indices (ETCCDI, ETCCDI (2018)) and meteorological drought indices such as the standardized precipitation index (SPI) and standardized precipitation evapotranspiration index (SPEI). Compound extreme events, known to carry disproportionate societal and economic costs, will also be a focus. Examples of compound extreme events under consideration include: heat stress (extreme temperature and humidity), wildfire conditions (extreme temperature and wind) and infrastructure damage (extreme rainfall and wind). We note that very few studies have comprehensively evalu-
ated compound extreme events in regional climate model ensembles to date. Moving beyond the simple summary/descriptive statistics of model evaluation is also a priority, with the intention to include more process-oriented diagnostics of model biases. Examples include interrogating why some models simulate extremes poorly as related to biases in surface turbulent fluxes in




the land surface model component (Kala et al., 2016) or biases in large scale atmospheric conditions (e.g. blocking) that can promote the onset of extreme events (Gibson et al., 2016).

Additionally, future work for RCMES will include the output of Bayesian metrics (or probabilistic metrics), such as those obtained by Bayesian Model Averaging (BMA, Raftery et al. (2005)) or Approximate Bayesian Computation (ABC). BMA
can directly provide an inter-model comparison by diagnosing the various models' abilities to mimic the observed data. Then a weight with associated uncertainty is assigned to each model based on its comparison to the observations, where higher weights indicate more trust-worthy models. Using an optimal combination of these weights, a more informed forecast/projection of the climate system can be made, which can potentially provide more accurate estimates of the impact on regional systems. Unfortunately, BMA utilizes a likelihood (or cost) function when determining the model weights, which can sometimes be
a roadblock for problems of high dimensions. Therefore, specified summary metrics can be defined for the problem, such as those related to extremes in precipitation or drought, and by using the ABC method we can replace the likelihood function needed for BMA with a cost function that minimizes only the difference between the observed and simulated metrics rather than differences between the entirety of the data. This likelihood-free type of estimation is attractive as it allows the user to disentangle information in a time- or space- domain and use it in a domain that may be more suitable for the regional analysis
(e.g. fitting the distribution of wintertime extreme precipitation or Summer time extreme temperature as possible metrics). By combining BMA with ABC, a diagnostic based approach for averaging regional climate models becomes possible.

The information technology component of RCMES will also have enhanced parallel processing capabilities. Our future development of OCW dataset processor will leverage the maturity and capabilities of open source libraries to facilitate the handling and distribution of massive datasets using parallel computing. Given the sizes of the multi-year model runs and
observations at high spatial and temporal resolutions (km and sub-hourly), scaling across a parallel computing cluster is needed to efficiently execute the analysis of fine-scale features. SciSpark is a parallel analytical engine for science data that uses the highly scalable MapReduce computing paradigm for in-memory computing on a cluster. SciSpark has been successfully applied to testing several RCMES use cases, such as Mesoscale Convective Systems (MCS) Characterization (Whitehall et al., 2015) and the probability density function clustering of surface temperature (Loikith et al., 2013). We will apply and test
SciSpark to analyses of high-resolution datasets and publish new versions of RCMES with parallel-capable examples.

Lastly, the development of RCMES aims to contribute to the CORDEX community and US NCA in order to enhance the visibility and utilization of NASA satellite observations in the community.

*Code and data availability.*  See Section 3.5 Download and installation of RCMES.

*Competing interests.*  The authors declare that we have no significant competing financial, professional or personal interests that might have
influenced the performance or presentation of the work described in this manuscript.





*Acknowledgements.* We acknowledge the Coordinated Regional Climate Downscaling Experiment under the auspices of the World Climate Research Programme's Working Group on Regional Climate. We would like to express our appreciation to Dr. William Gutowski and Dr. Linda Mearns for coordinating the overall CORDEX program and CORDEX North America respectively. The JPL authors' contribution to this work was performed at the Jet Propulsion Laboratory, California Institute of Technology, under contract with NASA.



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
