# Peer review of "Regional Climate Model Evaluation System powered by Apache Open Climate Workbench v1.3.0: an enabling tool for facilitating regional climate studies"

_Geoscientific Model Development, 2018_

## Referee Comment (RC1) · N. Massey (Referee) · 23 Jul 2018

General Comments

This paper describes a software suite, the Regional Climate Model Evaluation System (RCMES), which can be used to evaluate the performance of regional climate models, in particular those that are contributing to the CORDEX regional climate projections project. RCMES is built on top of the Open Climate Workbench (OCW), which is an open source and community-governed software toolkit to facilitate the analysis of climate models and observations. RCMES consists of two main parts - a database of observations for which to calibrate models against, and the RCMES toolkits. The toolkits provide functionality to load datasets in multiple formats and from four data sources, without having to know the URLs, etc. of the data sources. The toolkits also provide analysis routines for model evaluation as well as plotting routines for common plots such as the Taylor diagram.

From a software point of view, RCMES provides a valuable resource to the Regional Climate Modelling community in that it provides a standardised set of analysis tools which are comprehensive and easy to use. In particular, the three tiers of user interaction (the CLI, CFiles and using OCW with Python scripting) allows for a shallow learning curve. Standardised analysis routines enable analyses to be directly comparable across research groups, i.e. you know that the bias is computed in exactly the same way for each model / observation comparison. This is especially valuable to the CORDEX consortium, to enable comparison of the model analyses.

The community driven software development and community governance ensures that the software is freely available, that individual or teams of researches can make contributions but that code additions are approached in a peer-reviewed fashion to maintain quality.

The paper itself is comprehensive, well written and provides a valuable overview of the RCMES project, which will help users taking their first steps in using the software and provide a short reference and pointers to more in-depth resources for more experienced users.

I have few comments, mostly centred around grammar, typos, bad citations and clarifications.

I recommend publishing the paper with these minor corrections, which I have listed below.

[Figure]

"Specific comments"

P2L22: CMIP is currently in its sixth phase. The fifth phase has been completed, but is the one everyone is using. P7L2: Captialise "yaml" to "YAML". YAML is not part of Python, it is an independent mark up language, so remove "Python" before this P10L9: Can the loader use the CF-compliant standard name, from the attributes metadata in a netCDF file, to load the latitude, longitude, time and level variables? P10L20: Can the loader use OpenDAP to access a remote dataset? P11L2: what is a "granule" in this context? Please define. P12L22: Does the temporal resampling support sub 24-hour temporal resolution? e.g. 3 or 6 hourly. If not, does this mean the toolkit cannot be used to evaluate the diurnal cycle? Having a good representation of the diurnal cycle is very important for regional climate modelling, especially for heatwave extreme events. P14L2: by "slower" do you mean slower in installation time or slower in performance once installed? Please make clear. P14L20: you could also explore creating a Docker container, to make packaging the dependancies easy without the performance hit of a VM. P14L25: isn't github a "collaborative platform where climate scientists can spontaneously share their software updates". What is different here? Please explain. P15L16: "over the years": how long has OCW been active as a project. P18L16: "By combining BMA with ABC, a diagnostic based approach for averaging regional climate models becomes possible": please provide a reference for this.

"Technical corrections"

P2L7: comma after "Yet" P2L13: Change "Because of" to "Due to" P2L25: Change "based on GCM" to "which is based on GCM" P2L29: Change "is" to "are" in "is now underway" P2L32: Change "Because of" to "Due to" P3L8: Define "DOE" P3L8: Bad citation (/citepclimatemodeling) P3L35: Bad citation(citetpodaac) P4L2: change to "fostering the collaboration" or "fostering collaboration between" P4L5: put commas around "therefore" P4L7: capitalise python to Python P4L11: change "as well as describe" to "as well as to describe" P5L2: change "can utilize Open Climate Workbench" to "can utilize the Open Climate Workbench" P5L2: add "(OCW)" after "Open Climate

Workbench" P5L2: change "build up" to "write" or "produce" P4L7: capitalise python to Python P8L15: change "use" to "uses" in "RCMES CFiles use" P8L22: bad citation ("citetLee17") P10L2: capitalise python to Python P11L3: missing "a" between "covering" and "myriad" - i.e. should be "covering a myriad" P13L8: change "that are recently" to "that have been recently" P13L8: change "long-term trend simulated" to "long-term trend of simulated" P13L20: change "that are deviated from" to "that deviate from" P13L24: these downscaling method are not all simple! P13L26: capitalise "yaml" to "YAML" P14L4: capitalise "python" to "Python" P14L4: change "any types of" to "any type of" all "all types of" P14L6: capitalise "python" to "Python" P14L11: capitalise "python" to "Python" P15L3: bad citation citetapache P15L5: change "climate scientists" to "climate scientist" P15L6: change "becomes of" to "would become of" P15L6: "everyone and anyone" not necessary P16L3: by "committers" do you mean "code committers"? P16L6: change "last release" to "latest release" P16L8: change "is" to "are" in "contributions from any party is reviewed" P16L25: change "one which has" to "ones which have" P17L34: change "interrogating" to "determining". Comma after "poorly" P18L4: provide reference for "ABC" P18L6: commas around "with associated uncertainty" P18L17: change to "development of the OCW dataset processor" - add "the"

References error with references: "<GotoISI>" (undefined in LaTeX maybe?) please check all references

---

## Referee Comment (RC2) · Anonymous Referee #2 · 4 Aug 2018

This paper makes a review of The Regional Climate Model Evaluation System (RCMES) including basic components, features, usage, and installation.

**1   General comments:**

Even though the paper is a good review work of RCMES, I suggest to add some appendices for completion:

[Figure]

- Explained examples of CFiles (YAML).

- At least one (basic) run from start to end (including CFiles and outputs).

This information will help other scientists to understand and how to use the model better.

**2 Specific comments:**

- P8, l3: "... takes about 45 minutes on a multi-core Linux computing platform." Can you please detail the technical specs of the hardware as well as the Linux system (kernel, relevant libs...). This is important so other scientists can compare and decide what will be the cost of running RCMES on their own infrastructures.

- P8, l26-28: Given that some of the components are contained in another (e.g. RCMES database is a possible source for the Data Loader), and for readability reasons, I suggest to mention which section or subsection of the current document talks about each component.

- P11, l20: Please change 'json' to 'JSON'.

- P12, l2: Please change 'scipy' to 'SciPy'.

- P12, l24: I suggest to use the term 'function' instead of 'routine' on this case.

- P13, l30: "1) a map file, 2) histograms, and 3) a spreadsheet" Can you please detail the (file) formats available for this outputs?

- P14, l1: Do you provide any version for Python (any) Virtual Environments? If so, can you please detail it on this section?

- P14, l12: Can you please mention in this section that the VM image is for Virtu-alBox?

- P14, l14: Containers and VMs can be complementary technologies (depending on the case). I think for this context this phrase "As a completely self-contained package running in a virtual host container..." does not make sense at all. I suggest to either remove it or rephrase it with technically correct terms.

- P14, l16: "the latest Python" Unless you do continuous builds of the image I suggest to change this to just "Python" and include the major version.

---

## Referee Comment (RC3) · Anonymous Referee #3 · 17 Aug 2018

I must say this is the very first time since I started reviewing papers, that I truly do not have any comments/suggestions. I did read the paper and look carefully at the webpage of the RCMES (and found very useful the tutorial). So overall, I think the paper is very comprehensive and well written.

---

## Author Comment (AC1) · 14 Sep 2018

We would like to thank the three reviewers for providing valuable comments that helped in a better representation of our manuscript. Please find below our replies following the comments. Comments are listed first, followed by replies and associated changes. While revising the manuscript based on the comments, we have also corrected some errors throughout the manuscript to further improve it.

**Reviewer #1 (Dr. Massey)**

**General Comments**
This paper describes a software suite, the Regional Climate Model Evaluation System (RCMES), which can be used to evaluate the performance of regional climate models, in particular those that are contributing to the CORDEX regional climate projections project. RCMES is built on top of the Open Climate Workbench (OCW), which is an open source and community-governed software toolkit to facilitate the analysis of climate models and observations. RCMES consists of two main parts - a database of observations for which to calibrate models against, and the RCMES toolkits. The toolkits provide functionality to load datasets in multiple formats and from four data sources, without having to know the URLs, etc. of the data sources. The toolkits also provide analysis routines for model evaluation as well as plotting routines for common plots such as the Taylor diagram.

From a software point of view, RCMES provides a valuable resource to the Regional Climate Modelling community in that it provides a standardised set of analysis tools which are comprehensive and easy to use. In particular, the three tiers of user interaction (the CLI, CFiles and using OCW with Python scripting) allows for a shallow learning curve. Standardised analysis routines enable analyses to be directly comparable across research groups, i.e. you know that the bias is computed in exactly the same way for each model / observation comparison. This is especially valuable to the CORDEX consortium, to enable comparison of the model analyses.

The community driven software development and community governance ensures that the software is freely available, that individual or teams of researchers can make contributions but that code additions are approached in a peer-reviewed fashion to maintain quality.

The paper itself is comprehensive, well written and provides a valuable overview of the RCMES project, which will help users taking their first steps in using the software and provide a short reference and pointers to more in-depth resources for more experienced users.

I have few comments, mostly centred around grammar, typos, bad citations and clarifications.

I recommend publishing the paper with these minor corrections, which I have listed below.

→ We would like to thank Dr. Massey very much for providing valuable comments on the manuscript, especially those meticulous technical corrections.

**Specific comments**
P2L22: CMIP is currently in its sixth phase. The fifth phase has been completed, but is the one everyone is using.

→ We have corrected this.

P7L2: Captialise "yaml" to "YAML". YAML is not part of Python, it is an independent mark up language, so remove "Python" before this

→ We appreciate this comment. YAML in the manuscript has been capitalized.

P10L9: Can the loader use the CF-compliant standard name, from the attributes metadata in a netCDF file, to load the latitude, longitude, time and level variables?

→ Yes, it can. However, as stated in the following sentence, users can provide names for the latitude, longitude, and time to load non-standard files.

P10L20: Can the loader use OpenDAP to access a remote dataset?

→ The ESGF loader in the released version downloads a NetCDF file without subsetting the data using OpenDAP. When accessing PO.DAAC data, OpenDAP is used.

P11L2: what is a "granule" in this context? Please define.

→ The following sentence has been revised.
(Before) Current functionality includes the ability to retrieve and extract full granules and/or specific variables from over 50 Level 4 blended datasets covering myriad of gridded spatial resolutions between 0.05 and 0.25 degrees…
(After, P11L6) Granules are equal size subsets of a satellite's observations along its track. Current functionality includes the ability to retrieve and extract granules meeting a search criteria and/or specific variables from over 50 Level 4 blended datasets covering myriad of gridded spatial resolutions between 0.05 and 0.25 degrees…

P12L22: Does the temporal resampling support sub 24-hour temporal resolution? e.g. 3 or 6 hourly. If not, does this mean the toolkit cannot be used to evaluate the diurnal cycle? Having a good representation of the diurnal cycle is very important for regional climate modelling, especially for heatwave extreme events.

→ We thank the reviewer for providing this thoughtful comment. However, the resampling does not support sub-daily resolution. We will try to provide this functionality in the next release.

P14L2: by "slower" do you mean slower in installation time or slower in performance once installed? Please make clear.

→ The following sentence has been revised.
(Before) The latter is slower than the package installation, but it provides Linux OS and all required python libraries that can run on any types of users' computers.
(After, P14L12) RCMES execution with the latter is slower than the package installation, but it provides Linux OS and all required python libraries that can run on any types of users' computers.

P14L20: you could also explore creating a Docker container, to make packaging the dependancies easy without the performance hit of a VM.

→ We think this suggestion is very interesting. We are testing RCMES Docker containers to test Docker Swarm to parallelize RCM evaluations for multiple CORDEX domains. We plan to publish another paper on benchmarking different parallelization methods for RCMES.

P14L25: isn't github a "collaborative platform where climate scientists can spontaneously share their software updates". What is different here? Please explain.

→ Yes, github is a collaborative platform. However, many projects on github are not sustainable beyond the period of developers' funding. This is why we transitioned the RCMES development to the Apache Software Foundation's OCW.

P15L16: "over the years": how long has OCW been active as a project.
→ The following sentence has been revised.
(Before) In addition, over the years OCW has been presented in countless conferences.
(After, P15L25) In addition, OCW has been presented in countless conferences since its first release in June 2013.

P18L16: "By combining BMA with ABC, a diagnostic based approach for averaging regional climate models becomes possible": please provide a reference for this.
→ We have added Turner and Zandt (2012), Vrugt and Sadegh (2013), and Sadegh and Vrugt (2014) to the corresponding paragraph.

**Technical corrections**
P2L7:  comma after "Yet"
→ We have added a comma.

P2L13:  Change "Because of" to "Due to"
P2L25:  Change "based on GCM" to "which is based on GCM"
P2L29:  Change "is" to "are" in "is now underway"
P2L32: Change "Because of" to "Due to"
→ We have changed these as suggested. Thank you again.

P3L8: Define "DOE"
→ DOE stands for the United States Department of Energy. It is defined in the revised manuscript.

P3L8: Bad citation (/citepclimatemodeling)
P3L35:  Bad citation(citetpodaac)
P8L22: bad citation ("citetLee17")
P15L3: bad citation citetapache
→ We have corrected these and other bad citations.

P4L2:  change to "fostering the collaboration" or "fostering collaboration between"
→ We have changed this.

P4L5: put commas around "therefore"
(Before) To promote greater collaboration and participation of the climate research community within the RCMES development process, we transitioned from a closed-source development process to an open-source software (OSS) community driven project hosted in the public forum and therefore subject to public peer review, something which has significantly improved the overall project quality and standards the community and project holds itself to.

(After, P4L2) To promote greater collaboration and participation of the climate research community within the RCMES development process, we transitioned from a closed-source development process to an open-source software (OSS) community driven project hosted in the public forum. As a result, the development process is subject to public peer review, something which has significantly improved the overall project quality and standards the community and project holds itself to.

P4L7:  capitalise python to Python
P4L11:  change "as well as describe" to "as well as to describe"
P5L2: change "can utilize Open Climate Workbench" to "can utilize the Open Climate Workbench"
P5L2: add "(OCW)" after "Open Climate Workbench"
P5L2: change "build up" to "write" or "produce"
P10L2: capitalise python to Python
P11L3: missing "a" between "covering" and "myriad" - i.e. should be "covering a myriad"
P13L20: change "that are deviated from" to "that deviate from"
P13L26: capitalise "yaml" to "YAML"
P14L4: capitalise "python" to "Python"
P13L26: capitalise "yaml" to "YAML"
P14L4: capitalise "python" to "Python"
P14L6: capitalise "python" to "Python"
P14L11: capitalise "python" to "Python"
P15L5: change "climate scientists" to "climate scientist"
P15L6: change "becomes of" to "would become of"
P16L6: change "last release" to "latest release"
P16L8: change "is" to "are" in "contributions from any party is reviewed"
P18L6: commas around "with associated uncertainty"
P18L17: change to "development of the OCW dataset processor" - add "the"
→ We have applied all of these to the revised manuscript.

P8L15: change "use" to "uses" in "RCMES CFiles use"
→ CFiles stand for configuration files. So, we keep 'use'.

P13L8: change "that are recently" to "that have been recently"
P13L8: change "long-term trend simulated" to "long-term trend of simulated"
(Before) Another set of metrics that are recently added enables evaluation of long-term trends simulated climate models over the contiguous United States and analysis of the associated uncertainty.
(After, P13L12) Another set of metrics that have been recently added enables evaluation of long-term trends in climate models over the contiguous United States and analysis of the associated uncertainty.

P13L24: these downscaling method are not all simple!
→ We think that these four methods are relatively simple compared to state-of-art empirical downscaling techniques.

P14L4: change "any types of" to "any type of" all "all types of"

→ "any types of" has been replaced by "any type of".

P15L6: "everyone and anyone" not necessary

→ We agree with this. "everyone and anyone" has been replaced by "everyone".

P16L3: by "committers" do you mean "code committers"?

→ Yes. "committers" in P15 has been replaced by "code committers".

P16L25: change "one which has" to "ones which have"

→ We have changed "one which has" to "those which have".

P17L34: change "interrogating" to "determining". Comma after "poorly"

(Before) Examples include interrogating why some models simulate extremes poorly as related to biases

(After, P18L10) Examples include investigating why some models simulate extremes poorly, as related to biases

P18L4: provide reference for "ABC"

→ We have added Turner and Zandt (2012).

References error with references: "<GotoISI>" (undefined in LaTeX maybe?) please check all references

→ We have checked all references and corrected several errors. We appreciate the comment.

---

## Author Comment (AC2) · 14 Sep 2018

We would like to thank the three reviewers for providing valuable comments that helped in a better representation of our manuscript. Please find below our replies following the comments. Comments are listed first, followed by replies and associated changes. While revising the manuscript based on the comments, we have also corrected some errors throughout the manuscript to further improve it.

**Reviewer #2**

**General comments:**

Even though the paper is a good review work of RCMES, I suggest to add some appendices for completion:
• Explained examples of CFiles (YAML).
• At least one (basic) run from start to end (including CFiles and outputs).
This information will help other scientists to understand and how to use the model better.

→ We appreciate these comments. The RCMES website explains about CFiles (https://rcmes.jpl.nasa.gov/content/config-files) and describes basic example runs from start to end (https://rcmes.jpl.nasa.gov/content/tutorials-overview). CFiles and input datasets can be also downloaded. Please see P7L7 in the revised manuscript.
"The tutorials on the RCMES websites (JPL, 2018b) provide step-by-step instructions, CFiles, and datasets to reproduce all of the figures included in the two published articles."

**Specific comments:**
• P8, l3: "... takes about 45 minutes on a multi-core Linux computing platform." Can you please detail the technical specs of the hardware as well as the Linux system (kernel, relevant libs...). This is important so other scientists can compare and decide what will be the cost of running RCMES on their own infrastructures.

→ We have added some details about the hardware specs.
(Before) As an example, running RCMES for CORDEX North America domain with 12 variables and 3 seasons (36 unique evaluations with 5 datasets each) takes about 45 minutes on a multi-core Linux computing platform.
(After, P8L2) As an example, running RCMES for CORDEX North America domain with 12 variables and 3 seasons (36 unique evaluations with 5 datasets each) takes about 45 minutes using an Intel Xeon CPU with a clock rate of 2.30GHz on a multi-core Linux computing platform.

• P8, l26-28: Given that some of the components are contained in another (e.g. RCMES database is a possible source for the Data Loader), and for readability reasons, I suggest to mention which section or subsection of the current document talks about each component.

→ We have revised the following paragraph.
(Before) In the following, we describe seven software components of RCMES, 1) data loader, 2) the RCMES database, 3) dataset processor, 4) metrics, and 5) plotter, 6) statistical downscaling module, and 7) installation package options for disseminating RCMES.
(After, P8L27) In the following, we describe seven software components of RCMES, 1) data loader (Section 3.1), 2) the RCMES database (Section 3.1.1), 3) dataset processor (Section 3.2), 4)

metrics and 5) plotter (Section 3.3), 6) statistical downscaling module (Section 3.4), and 7) installation package options for disseminating RCMES (Section 3.5).

- P11, l20: Please change 'json' to 'JSON'.
- P12, l2: Please change 'scipy' to 'SciPy'.
- P12, l24: I suggest to use the term 'function' instead of 'routine' on this case.

→ We have changed these as suggested and 'numpy' to 'NumPy'.

- P13, l30: "1) a map file, 2) histograms, and 3) a spreadsheet" Can you please detail the (file) formats available for this outputs?

→ We have added the following sentence.
(Added, P14L6) The map and histogram are portable graphics format (PNG) files, and the spreadsheet file has the eXceL Spreadsheet (XLS) extension.

- P14, l1: Do you provide any version for Python (any) Virtual Environments? If so, can you please detail it on this section?
- P14, l16: "the latest Python" Unless you do continuous builds of the image I suggest to change this to just "Python" and include the major version.

→ The VM includes Python 2.7. We have added the version information.

- P14, l12: Can you please mention in this section that the VM image is for VirtualBox?
- P14, l14: Containers and VMs can be complementary technologies (depending on the case). I think for this context this phrase "As a completely self-contained package running in a virtual host container..." does not make sense at all. I suggest to either remove it or rephrase it with technically correct terms.

→ We agree with this. The following sentence has been revised.
(Before) As a completely self-contained package running in a virtual host container, the VM approach offers a plug and play approach that enables users to quickly begin exploring RCMES without any trouble in the installation process.
(After, P14L24) As a completely self-contained research environment running with VirtualBox \citep{virtualbox}, the VM approach offers a plug and play approach that enables users to quickly begin exploring RCMES without any trouble in the installation process.

---

## Author Comment (AC3) · 14 Sep 2018

We would like to thank the reviewer. We have improved the manuscript based on the comments from the other two reviewers.

---

## Author Comment (AC4) · 14 Sep 2018

The comment was uploaded in the form of a supplement:
https://www.geosci-model-dev-discuss.net/gmd-2018-113/gmd-2018-113-AC4-supplement.zip

---

## Author Response (AR2)

We would like to thank Dr. Juan Antonio Añel for valuable comments. As indicated in the following responses, we have incorporated all these comments into our new revision.

**Comments to the Author:**

You have addressed most of the comments by the reviewers but some improvements to your manuscript are necessary before it is publishable. More specifically:

1. there are several errors in citations/references, probably as a consequence of a wrong use of BibTeX to build the list of references:
* page 3, line 10 should say (DOE, 2018) not (of Energy, 2018)
* page 4, line 3: PO.DAAC is repeated
* page 15, line 12: (ASF, 2018a)
→ We have corrected the errors.

2. page 7, line 6: I guess you mean "GNU/Linux". I do not think that you are using only a kernel without any extra layer of software to run your computers. Please, correct it.
→ We have corrected this. Thank you again for pointing out this.

3. page 14: it is quite a shame that the output format for spreadsheets is .XLS. I understand that nothing prevents the output of being in OpenDocument Format (.ODS), a fully compatible ISO standard for documentation that any other XLS compatible software that you have in mind can open. However the opposite case does not apply. Moreover you do not mention what XLS implementation use. So, would it be a big problem to get an output in .ODS and if not could you clarify the .XLS standard used for it?;
→ We thank the editor for this comment. RCMES users have not had any issue of the file format as they can open the output file using Open Office or Microsoft Excel. However, we agree that .ODS format is much better than .XLS in terms of compatibility. We will definitely consider this in a future release of RCMES and OCW.

4. Section 4:
* at the beginning of the section you talk about "software releases protected by appropriate software license(s)": this is wrong. Licenses do not protect software. Licenses establish the terms of use and the owner of the intellectual property. Only a few legislations let to "protect" (better to say that they restrict use) software and this is done through patents (e.g. the USA), a completely different thing to a license. So such statement is wrong. I do not know what the authors of the software pretended but it is wrongly exposed.
* also the next statement is wrong: "these activities require...peer review". To release the code does not require refactoring, testing or documentation. To release code is easy, straightforward and it is worthy at any stage of the development process.
https://cacm.acm.org/magazines/2011/5/107698-the-importance-of-reviewing-the-code
A different issue is that you get embarrassed by what you release or that it is not working at all for others, in this case the appropriate sentence would be "in order to do not get embarrassed

by the poor quality of the development process extra effort is necessary...". And probably stating such thing is not adding anything here. So better, simply remove the sentence on the need of extra effort;

* next sentence: "However, it is not easy... ". I do not agree with you at all. There are a lot of free platforms (both from the point of view of freedom and money) to develop code in a collaborative way. It is really easy to find one: Github, Gitlab, Savannah, SourceForge,... If you mean that part of the scientific community is reluctant to use them, this is a sociological issue and it does not have nothing to do with your work. Moreover you do not need it in order to simply state that you have used Apache as your choice. So please, remove it.

→ We really appreciate these comments. The following paragraph has been revised.

(Before)

[revised manuscript text omitted]